

# Comparative analyses of chloroplast genomes in 'Red Fuji' apples: low rate of chloroplast genome mutations

Haoyu Miao[*], Jinbo Bao[*], Xueli Li, Zhijie Ding and Xinmin Tian

Xinjiang Key Laboratory of Biological Resources and Genetic Engineering, College of Life Science and Technology, Xinjiang University, Xinjiang, Urumqi, China
[*] These authors contributed equally to this work.

## ABSTRACT

**Background**. Fuji is a vital apple cultivar, and has been propagated clonally for nearly a century. The chloroplast genome variation of Fuji apples in China has not been investigated.

**Methods**. This study used next-generation high-throughput sequencing and bioinformatics to compare and analyze the chloroplast genome of 24 Red Fuji varieties from nine regions in China.

**Results**. The results showed that the 24 chloroplast genomes were highly conserved in genome size, structure, and organization. The length of the genomes ranged from 160,063 to 160,070 bp, and the GC content was 36.6%. Each of the 24 chloroplast genomes encoded 131 genes, including 84 protein-coding genes, 37 tRNA genes, and eight rRNA genes. The results of repeat sequence detection were consistent; the most common sequence was forward repeats (53.1%), and the least common sequence was complementary repeats (4.1%). The chloroplast genome sequence of Red Fuji was highly conserved. Two indels were detected, but the PI value was 0, and there were no SNP loci. The chloroplast genome variation rate of Red Fuji was low.

## INTRODUCTION

Apple (*Malus × domestica*) belongs to the genus *Malus* in the family Rosaceae. It is among the most popular and nutritious fruits (*Jung & Main, 2014*), In 2018, China, which is the largest producer of apples in the world (*Li et al., 2020*), produced more than half (43.88 million tons) of the global production (80.82 million tons). In addition to being used as fresh, natural products, apples can also be processed into foods, like apple juice, applesauce, and dried fruits (*Deng & Li, 2020*). Apple cultivation has a significant value and significance in China's economic development. Many excellent varieties, such as 'Fuji' have been widely cultivated (*Kuang et al., 2020*). 'Fuji' was generated in Japan from '*Ralls Janet*' ×'*Delicious*' (*Hummer & Janick, 2009*). 'Fuji' apple is one of China's most popular late-ripening varieties, accounting for 72.7% of the total Chinese apple production (*An et al., 2018*). Fuji has many advantages, such as colorfulness, sweet taste, storage resistance, and high profit. It is sweeter and crisper than many other apple varieties, and

Corresponding author
Xinmin Tian, tianxm06@lzu.edu.cn

has a longer shelf-life (*Li, Sun & Wang, 2019*). It is widely consumed worldwide due to its excellent properties (*He & Zhang, 2020*). The economic value of 'Fuji' apples, highlights the importance of research on breeding and crop improvement. Most of the research on Fuji apples has concentrated on the nuclear genome. Research on organellar genomes is scarce. The non-nuclear genome includes the chloroplast (cp) and the mitochondrial genomes.

Chloroplasts have an independent genetic system and are semi-autonomous organelles (*Sugiura, 2003*). In 1978, the first chloroplast gene was isolated using molecular cloning by *Rochaix (1978)*. Some years later, *Ohyama et al. (1986)* and *Shinozaki et al. (1986)* sequenced the whole cp genome of *Nicotiana tabacum* and *Marchantia polymorpha*. For several reasons, the cp genome is more commonly used in molecular evolution and phylogenetic studies than the mitochondrial genome (*Guo et al., 2021*). Firstly, the cp genome includes a large amount of genetic information, which provides a database for comparative research (*Zuo et al., 2017*). Secondly, the nucleotide substitution rate is moderate and valuable in evolutionary studies (*Moore et al., 2010*). The molecular evolutionary rates in various regions of the cp genome are significantly different and could be used for different levels of systematic research (*Li et al., 2018*). In addition, the moderate cp genome size is easy to sequence, and there is good collinearity between the cp genomes of various plant groups, making it convenient for comparative analysis (*Alzahrani, 2021*). Therefore, the cp genome has a significant impact on the development of phylogenetic genomics (*Wang et al., 2021*). The number of cp genomes that have been sequenced has increased exponentially owing to the development of sequencing technology (*Liu et al., 2012*) The cp genome has been successfully used to analyze the phylogenetic relationships between numerous difficult species and study the structural characteristics, variation, and evolution of plants (*Yagi & Shiina, 2014*). For example, *Jansen (2007)* used 64 cp genomes to determine the phylogenetic relationship between angiosperms. Moreover, through homology analysis of the cp genome, we can deal with crop origin and domestication's important scientific problems and detect the changes in crop genome structure and gene sequence during domestication (*Abdullah et al., 2020*).

*Yan et al. (2019)* assembled and annotated the cp genome of cultivated apple (Yantai Fuji 8), which enriched the potential genetic resources for apple breeding. The quality of the same apple species can vary greatly in different plantations because of climate (*Fu et al., 2013*). Genetic changes of the genome can occur rapidly (*Jiao et al., 2014*). After nearly a century of clonal propagation, it is unknown whether the cp DNA of Red Fuji apples planted in different areas differs and whether there is a variation. This study, characterized the complete cp genome sequence of 24 Red Fuji accessions in nine regions and carried out a structural variation analysis. This study aimed to investigate mutations in the chloroplast DNA of the Red Fuji apple.

## MATERIALS & METHODS

### Plant materials

Germplasm of 24 Red Fuji apples was collected from nine regions in China. Two or three accessions were selected from each region for repeated experiments. The sources and

**Table 1 Germplasm resources of Red Fuji apple in nine regions.**

| | Region | Amount | GenBank numbers |
|---|---|---|---|
| 1 | Wunan Town, Wuwei City | 2 | OK458680, OK585094 |
| 2 | Qingyuan Town, Wuwei City | 2 | OK514184, OK585095 |
| 3 | Shanxi Province | 3 | OK585096, OK514185, OK585097 |
| 4 | Aksu | 3 | OK514186, OK585098, OK585099 |
| 5 | Tianshui City | 3 | OK585100, OK514187, OK585101 |
| 6 | Li County, Longnan | 3 | OK514188, OK585102, OK585103 |
| 7 | Jiuquan city | 3 | OK514189, OK585104, OK585105 |
| 8 | Xingcheng City, Liaoning Province | 2 | OK514190, OK585106 |
| 9 | Zhengzhou Fruit Tree Institute | 3 | OK585107, OK514191, OK585108 |

quantities of the accessions are listed in Table 1. Fresh leaves of Red Fuji were dried in silica gel and taken to the laboratory.

## DNA extraction and sequencing

Total genomic DNA of each material was extracted using the plant genomic DNA secure Kit (DP320) of China Tiangen Biotechnology (Beijing) Co., Ltd. The integrity and purity of the DNA samples were determined by agarose gel electrophoresis. NuoheZhiyuan Technology Co., Ltd. (Beijing, China) used the Illumina Hiseq 2000 sequencing platform to perform genome sequencing of quality DNA samples. The read length was 150 bp. A total of 492.617 G of raw data was generated by sequencing, and the filtered clean data was 491.884 G, with an average of 20987.043M per sample (Table S1).

## Plastome assembly and annotation

We chose a reference map for assembly. The obtained clean data were assembled using Novoplasty software (https://github.com/ndierckx/NOVOPlasty), the configuration file config.txt needs to be set before use. Download the published chloroplast genome of Malus yunnanensis (GB: MH394390) from NCBI as the reference file, select its *rbc* L gene as the seed file, kmer = 39. The preliminary annotation of the assembled plastome was performed using Geseq (https://chlorobox.mpimp-golm.mpg.de/geseq.html). Then, Sequin 16.0 was used to correct the start and stop codons and intron/exon boundaries for the genes with software annotation errors. Finally, the complete cp genome sequence and its annotations were submitted to GenBank. The specific serial numbers are detailed in Table 1.

## Characterization and comparative analyses of chloroplast genomes

The total length of the genome, the length of each region (large single-copy regions, small single-copy regions, inverted repeat), gene composition, base composition and GC (AT) content were calculated using Geneious 11.1.2 software, The characteristics of Red Fuji genomes were analyzed.

## Boundary regions and comparative analysis

Different plant species have different gene sequences in the four junction regions. The change of cp genome length is the main reason for the contraction and expansion of the

IR region (*Raubeson et al., 2007*). The border regions between large single copy (LSC) and IR regions, and between small single copy (SSC) and IR regions, were compared using IRscope (https://irscope.shinyapps.io/irapp/).

## Codon usage analyses

Codon usage bias (CUB) refers to the phenomenon that some codons are used more than other synonymous codons in the process of gene translation between different species or within the same species (*Alexandra & Tamir, 2014*). CUB is a useful tool for understanding genetic and evolutionary processes. In this study, MEGA7.0 software CodonW was used to analyze codon preference, The results of the analysis are shown in charts.

## Repeat sequence and SSRs analyses

The online REPuters (https://bibiserv.cebitec.uni-bielefeld.de/reputer/) software was used to identify repeats which contain forward, reverse, and complex, repeats as well as palindromes of the cp genome of Red Fuji (*Frazer et al., 2001*). The following parameters were used to identify repeats with REPuter: Hamming distance of 3, Maximum Computed Repeats of 50, and repeat size >30 bp. Simple sequence repeats (SSRs) were checked using MISA (https://webblast.ipk-gatersleben.de/misa/index.php), with motif sizes of 1–6 nucleotides (*Thiel et al., 2003*). All other parameters were used as default values. The parameters of repetitive units and the minimum number of repetitions were set to 10 repetitions for single nucleotide type, 6 repetitions for dinucleotide type, 5 repetitions for trinucleotide type, 5 repetitions for tetranucleotide type, 5 repetitions for pentanucleotide type, and 5 repetitions for hexanucleotide type.

## Genome comparison

The cp genomes of Red Fuji apples from nine regions were compared and visualized by mVISTA (http://genome.lbl.gov/vista/index.shtml) online software. mVISTA is an online tool used for multiple DNA sequence alignments, where sequence similarity can be evaluated by comparing coding regions with non-coding regions, introns, and exons (*Frazer et al., 2004*). The genome of *M. sieversii* was selected as the reference, and the input file was the original FASTA format nucleotide sequence file and the gff3 format annotation file. The nucleotide diversity (PI) of Red Fuji was calculated in Dnaspv6 software. SNP and indel detection was carried out using Geneious 11.1.2.

## Phylogenetic analysis

A phylogenetic tree was constructed based on the combined cp genomes. *Pyrus pyrifolia* was used as the outgroup. The species and the accession numbers of their cp genomes in NCBI are listed in Table S2. The sequences were aligned using Mafft (https://mafft.cbrc.jp/alignment/server/). The model is GTRGAMMA+I. Maximum Likelihood (ML) methods were used to construct phylogenetic trees was used to perform the RAxML-HPC BlackBox 8.2.12 in CIPIES (https://www.phylo.org/) (*Salichos, Stamatakis & Rokas, 2014*) the ML analysis.
## RESULTS

### Characteristics of Red Fuji cp genomes

The 24 cp genomes of the Red Fuji apples collected were highly conserved in terms of gene content, gene order, and gene intron number. The genomes ranged in length from 160,063 bp to 160,070 bp, and as expected, the cp genomes contained the LSC and SSC, separated by a pair of IR regions (Fig. 1). The length of the IR region was 26,307–26,308 bp, the LSC region was 88,272–88,274bp, and the SSC region was 19,176–19,181 bp. The overall GC content was approximately 36.6%. The GC content in the IR region was 42.7%, which was higher than 34.2% in the LSC region and 30.4% in the SSC region (Tables 2 and 3). These numerical values were consistent among the 24 accessions and did not change.

Each plastome encoded 131 genes, including 84 protein-coding genes (PCGs), 37 transfer RNA (tRNA) genes, and eight ribosomal RNA (rRNA) genes. There were 22 introns in the annotated genes. Of the annotated genes, 18 embodied one intron, and two (*ycf3* and *clpP*) had two introns. Additionally, 62 protein-coding and 22 tRNA genes were located within the LSC; 12 protein-coding genes, 14 tRNA coding genes, and 8 rRNA coding genes were located within IRs, and 12 protein-coding genes and one tRNA gene were located within SSC. The genes in the LSC region accounted for 64.1% of the cp genome, the two IR regions accounted for 26%, and the genes in the SSC region accounted for 9.9%.

### IR boundary analysis

First, IR analysis was performed on the materials from each region, and the results showed that the cp genomes of Red Fuji in the same region were highly consistent (Figs. S1–S9). Therefore, the subsequent analysis was based on comparing the nine regions Nine samples were selected from each region to compare the distribution of genes in the boundaries of the four regions of the cp genome. The IR regions of the nine Fuji genomes sequenced were highly conserved (Fig. S10). We compared gene variation at the boundaries of five plastosomes of three Red Fuji, one *M. sieversii* (MK434920), and one *M. sylvestris* (MK434924). The results displayed that the boundaries of the four regions were relatively conserved, and the gene types distributed in the boundary region were highly consistent. The genes distributed in the boundary between the LSC and IR region were *rps* 19 and *trn* H-GUG. In addition, *rp1* 2, *ycf* 1, and *ndh* F are distributed in the boundary of the SSC/IR region. (Fig. 2). The length of *ycf* 1 extending to the IR region was 1,074 bp, and that of *ndh* F was 12 bp. The length of *rps* 19 extending to the IRb region extended was 69 or 115 bp, depending on the different species. The position of the gene in the boundary region was relatively fixed. There were no differences in the boundaries of the five samples. Hence, these results indicate that the IR region is highly conserved.

### Codon usage analyses

Since the base composition and AT/GC content was the same in the 24 genomes of Red Fuji, we chose one of them to calculate the frequency of amino acid and codon usage. The codon usage results of materials from Wunan Town, Wuwei City showed that a total of 20 amino acids (excluding the stop codon) were encoded, and the usage frequency of each amino acid ranged from 1.56% to 9.38%. Leucine, serine, and arginine were the most abundant amino
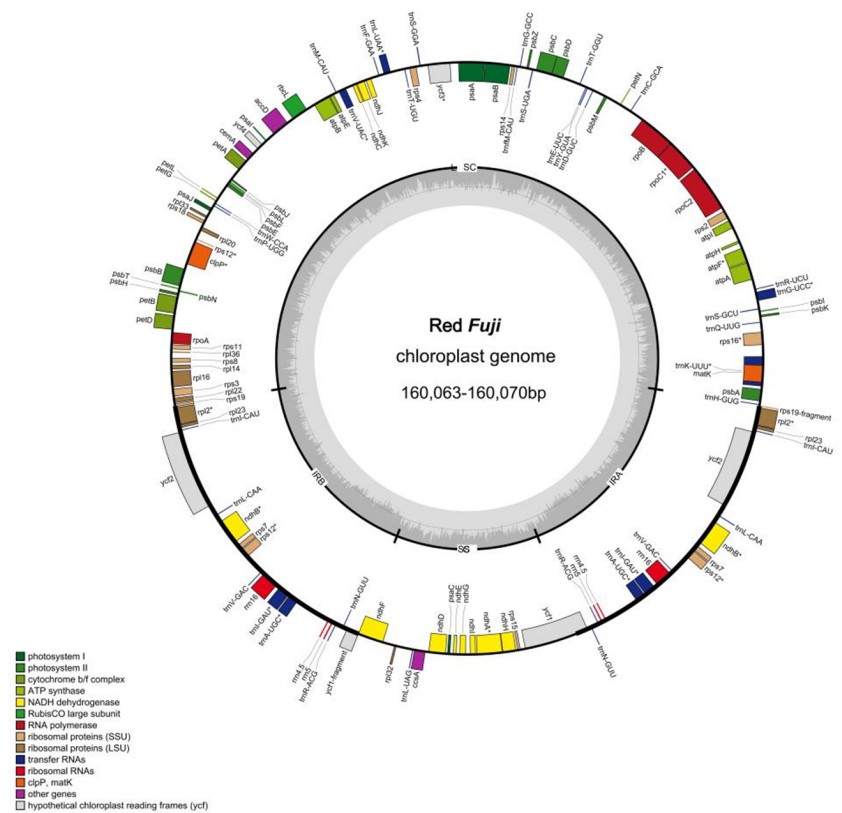

**Figure 1 Genome map of Red Fuji chloroplast.** Different colors represent different genes. The inner genesare transcribed clockwise, the outer genesare transcribed counterclockwise. The innermost gray pattern represents GC content.

**Table 2 Statistics of the chloroplast genomes of Red Fuji.**

|  | Size (bp) | LSC (bp) | SSC (bp) | IR (bp) | protein-coding genes | tRNA genes | rRNA genes | GC content(%) |
|---|---|---|---|---|---|---|---|---|
| WW | 160,068 | 88,273 | 19,181 | 26,307 | 84 | 37 | 8 | 36.6 |
| WQ | 160,069 | 88,274 | 19,181 | 26,307 | 84 | 37 | 8 | 36.6 |
| SX | 160,069 | 88,274 | 19,181 | 26,307 | 84 | 37 | 8 | 36.6 |
| AKS | 160,068 | 88,273 | 19,181 | 26,307 | 84 | 37 | 8 | 36.6 |
| ZZ | 160,070 | 88,274 | 19,181 | 26,308 | 84 | 37 | 8 | 36.6 |
| TS | 160,069 | 88,274 | 19,181 | 26,307 | 84 | 37 | 8 | 36.6 |
| LL | 160,066 | 88,272 | 19,180 | 26,307 | 84 | 37 | 8 | 36.6 |
| LX | 160,063 | 88,273 | 19,176 | 26,307 | 84 | 37 | 8 | 36.6 |
| JQ | 160,067 | 88,272 | 19,181 | 26,907 | 84 | 37 | 8 | 36.6 |

**Notes.**

WW, Wunan Town, Wuwei City; WQ, Qingyuan Town, Wuwei City; SX, Shanxi Province; AKS, Aksu; ZZ, Zhengzhou Fruit Tree Institute; TS, TianshuiCity; LL, Li County, Longnan; JQ, jiuquan city; LX, Xingcheng City, Liaoning Province.

**Table 3  Base composition of chloroplast genome of Red Fuji apple in different regions.**

|  |  | WW | WQ | SX | AKS | ZZ | TS | LL | LX | JQ |
|---|---|---|---|---|---|---|---|---|---|---|
| LSC(%) | A | 32.2 | 32.2 | 32.2 | 32.2 | 32.2 | 32.2 | 32.2 | 32.2 | 32.2 |
|  | C | 17.6 | 17.6 | 17.6 | 17.6 | 17.6 | 17.6 | 17.6 | 17.6 | 17.6 |
|  | G | 16.6 | 16.6 | 16.6 | 16.6 | 16.6 | 16.6 | 16.6 | 16.6 | 16.6 |
|  | T | 33.6 | 33.6 | 33.6 | 33.6 | 33.6 | 33.6 | 33.6 | 33.6 | 33.6 |
|  | GC | 34.2 | 34.2 | 34.2 | 34.2 | 34.2 | 34.2 | 34.2 | 34.2 | 34.2 |
| SSC(%) | A | 34.6 | 34.7 | 34.7 | 34.7 | 34.7 | 34.7 | 34.7 | 34.7 | 34.7 |
|  | C | 15.9 | 15.9 | 15.9 | 15.9 | 15.9 | 15.9 | 15.9 | 15.9 | 15.9 |
|  | G | 14.5 | 14.5 | 14.5 | 14.5 | 14.5 | 14.5 | 14.5 | 14.5 | 14.5 |
|  | T | 34.8 | 34.8 | 34.8 | 34.8 | 34.8 | 34.8 | 34.8 | 34.8 | 34.8 |
|  | GC | 30.4 | 30.4 | 30.4 | 30.4 | 30.4 | 30.4 | 30.4 | 30.4 | 30.4 |
| IRa(%) | A | 28.5 | 28.5 | 28.5 | 28.5 | 28.5 | 28.5 | 28.5 | 28.5 | 28.5 |
|  | C | 22.1 | 22.1 | 22.1 | 22.1 | 22.1 | 22.1 | 22.1 | 22.1 | 22.1 |
|  | G | 20.6 | 20.6 | 20.6 | 20.6 | 20.6 | 20.6 | 20.6 | 20.6 | 20.6 |
|  | T | 28.8 | 28.8 | 28.8 | 28.8 | 28.8 | 28.8 | 28.8 | 28.8 | 28.8 |
|  | GC | 42.7 | 42.7 | 42.7 | 42.7 | 42.7 | 42.7 | 42.7 | 42.7 | 42.7 |
| Total(%) | A | 24.3 | 24.3 | 24.3 | 24.3 | 24.3 | 24.3 | 24.3 | 24.3 | 24.3 |
|  | C | 18.6 | 18.6 | 18.6 | 18.6 | 18.6 | 18.6 | 18.6 | 18.6 | 18.6 |
|  | G | 17.9 | 17.9 | 17.9 | 17.9 | 17.9 | 17.9 | 17.9 | 17.9 | 17.9 |
|  | T | 32.1 | 32.1 | 32.1 | 32.1 | 32.1 | 32.1 | 32.1 | 32.1 | 32.1 |
|  | GC | 36.6 | 36.6 | 36.6 | 36.6 | 36.6 | 36.6 | 36.6 | 36.6 | 36.6 |

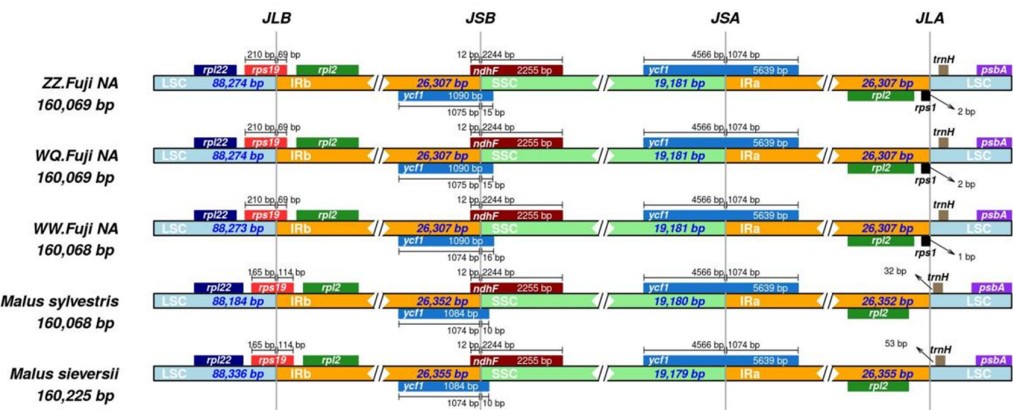

**Figure 2  Comparison of chloroplast genome boundary regions in Red Fuji, *M. sieversii* and *M. sylvestris*.**

acids. Among the encoded amino acids, except methionine and tryptophan, other amino acids were encoded by two to six synonymous codons (Fig. 3). For example, serine, arginine, and leucine were encoded by six synonymous codons, of which the most frequently used codons were TTA, TCT, and CGT, respectively. Four synonymous codons were used for

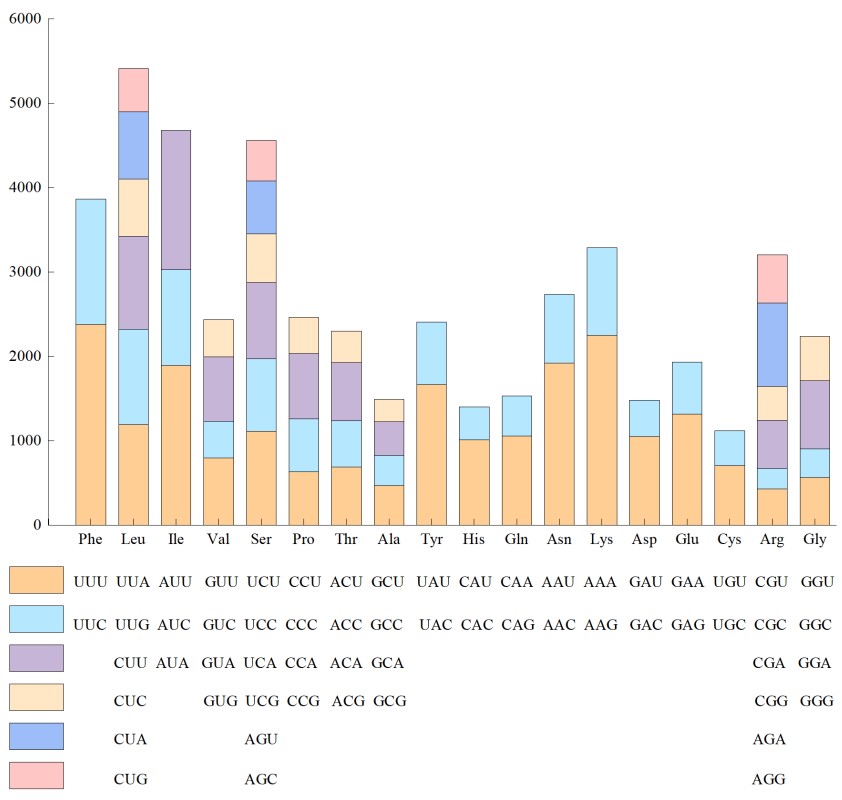

**Figure 3** Codon content of amino acids encoding proteins in the chloroplast genome of Red Fuji.

valine, proline, threonine, alanine, and glycine. The most frequently used codons were GTT, CCT, ACT, GCT, and GGT, respectively. Three synonymous codons were used for isoleucine, of which ATT was the most frequently used codon. Two synonymous codons were used for phenylalanine, tyrosine, histidine, glutamine, asparagine, lysine, aspartic acid, glutamic acid, and cysteine, which had different frequencies of use. The codons with a higher frequency for these nine amino acids were TTT, TAT, CAT, CAA, AAT, AAA, GAT, GAA, and TGT, respectively. The cp genome of Red Fuji apple has 32 codons with RSCU >1, of which 30 codons end with A/T, accounting for 93.75%. Most of the 29 codons with relative synonymous codon usage values (RSCU)<1 ended in G/C, accounting for 96.55%. Our results were similar to those of other angiosperms *Qian et al. (2013)*. The codons of the cp genome of Red Fuji that were used frequently at the third codon positions were A and T.

## Repeats and microsatellites analyses

A total of 49 repeat sequences were detected in each material, which consisted of 26 forward, 17 palindromic, four reverse, and two complement repeats (Table S3). Among them, there are 33 repeats with a length of 30–39 bp, 6 repeats with a length of 40–49 bp, and two repeats with more than 50 bp (Fig. 4). The most common type was forward repeats (53.1%), and the least common was complement-type repeats (4.1%). Using the

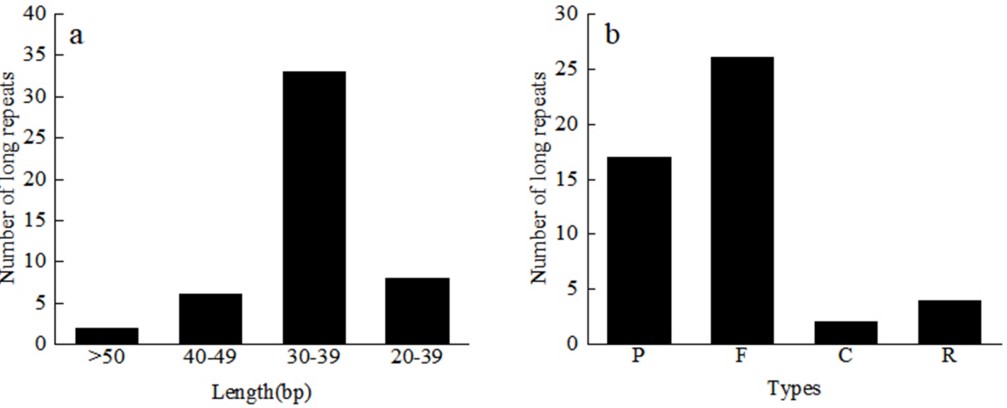

**Figure 4  Length and number of different types of long repeats.** C, Complementary-type repetition; F, Forward repetition; P, Palindrome repetition; R, Reverse repetition.

**Table 4  Statistics of simple repeats (SSRs) of chloroplast genome.**

|  | WW | WQ | SX | AKS | TS | LL | JQ | LX | ZZ |
|---|---|---|---|---|---|---|---|---|---|
| SSR | 57 | 58 | 58 | 57 | 58 | 57 | 57 | 57 | 58 |
| P1 | 46 | 48 | 47 | 47 | 47 | 47 | 46 | 46 | 47 |
| P2 | 2 | 2 | 2 | 2 | 2 | 2 | 2 | 2 | 2 |
| Pc | 9 | 9 | 9 | 9 | 9 | 8 | 9 | 9 | 9 |
| LSC | 46 | 47 | 47 | 46 | 47 | 46 | 46 | 46 | 47 |
| SSC | 5 | 5 | 5 | 5 | 5 | 5 | 5 | 5 | 5 |
| IR | 6 | 6 | 6 | 6 | 6 | 6 | 6 | 6 | 6 |

**Notes.**
P1: mononucleotide. P2: Dinucleotide repeat. Pc: complex polynucleotides.

MISA software, we searched for the SSR loci in the 24 Red Fuji apple cp genomes. A total of 1,380 microsatellites, two dinucleotide repeats, 46 to 48 mononucleotides repeats, and 8 or 9 complex polynucleotides were found. Mononucleotides (81.2%), mainly poly-A (polyadenine) and poly-T (polythymine), formed the largest proportion of SSRs in the Red Fuji cp genome. The majority of SSRs were situated in the intergenic region of LSC (Table 4).

## Genome comparison and divergence analyses

In this study, the online software mVISTA was used for sequence alignment and variation analysis of cp genome of Red Fuji, using *M. sieversii* (MK434920) as the reference. The nucleotide sequence similarity of the ten cp genomes was extremely high, suggesting that there was no variation in the cp genome of Red Fuji compared with its ancestral species (Fig. 5). At the same time, it can be found that divergence existed in the highly conserve dregions, and the coding region and IR region were more conserved. The intergenic regions with the highest levels of divergence were those in between *psb* L-*atp* A, *psb* M-*psb* D, and *ndh* c-*atp* E.

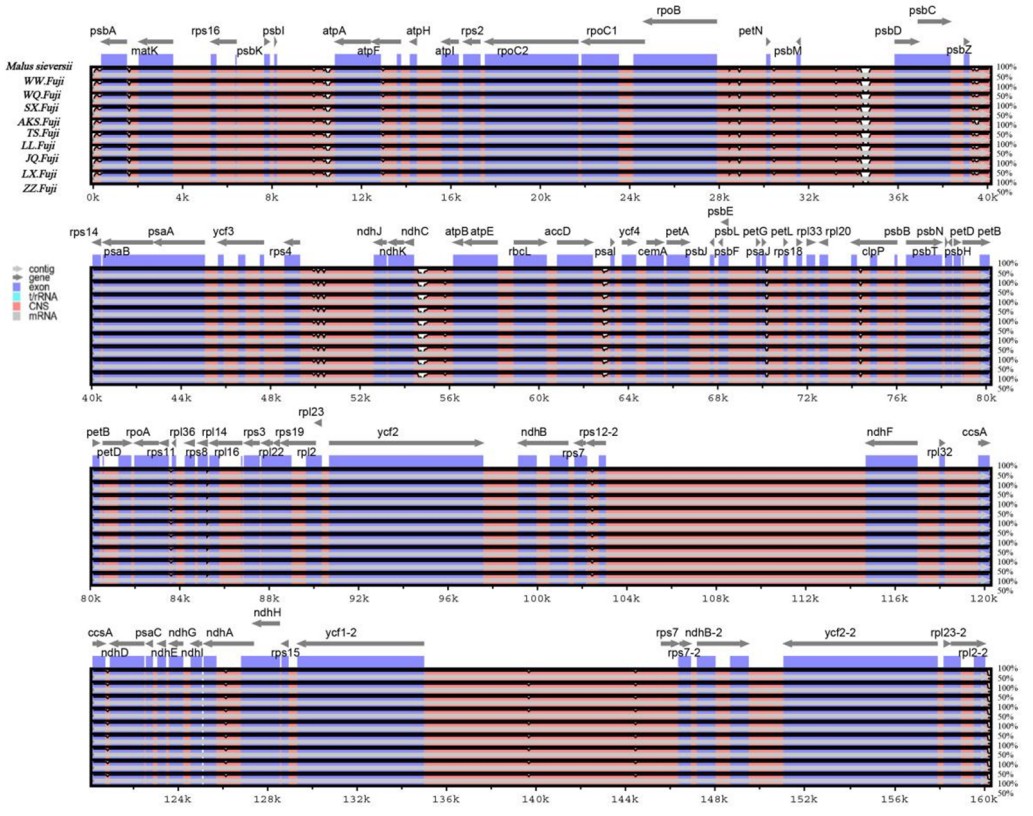

**Figure 5  Sequence alignment of nine Red Fuji chloroplast genomes using the mVISTA software with *M. sieversii* as a reference.** The *y*-axis represents the percent identity within 50–100%. The transcriptional direction of genes is indicated by grey arrows. Genome regions are color-coded as protein-coding (exon), tRNA, rRNA, and conserved non-coding sequences (CNS).

The nucleotide polymorphisms were analyzed using the DNAsp software, and the nucleotide diversity (PI) was 0. Geneious was used to detect nine regions' single nucleotide polymorphisms (SNPs) of *M. sieversii*, *M. sylvestris*, and Red Fuji apples in nine regions, Only two indel loci were detected in the *rps* 16 and *ndh* A genes. The results suggested that the rate of variation of Red Fuji in different regions was rarely low.

### Phylogenetic analysis

Phylogenetic analysis using Maximum likelihood (ML) yielded well-supported tree topologies (Fig. 6). All red Fuji are clustered into one branch, *M. sieversii* and *M. sylvestris*, also clustered into a branch, that diverged earlier than Red Fuji, suggesting that they are the ancestors of the cultivated species.

## DISCUSSION

This study compared the cp genome sequences of 24 Red Fuji apples The genomes were very similar to those of other *Malus* species regarding genome size, gene content, gene sequence, and GC content (*Bao et al., 2016*). The length of all the cp genomes was very

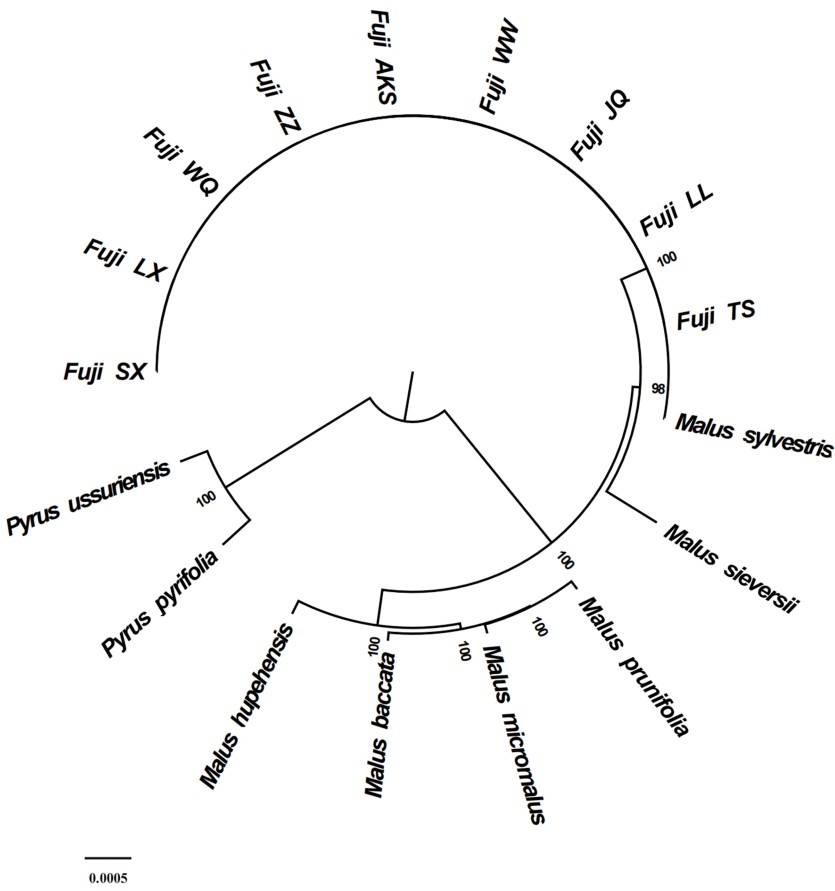

**Figure 6** Phylogenetic tree reconstruction using maximum likelihood.

similar, ranging from 160,063 to 160,070 bp, t The length of the cp genomes of *M. sieversii* is 160,025 bp, and that of *M. sylvestris* 160,068 bp. These are slightly different from that of Red Fuji. All the genomes encoded 131 genes, which were predicted to be protein-coding genes (84), tRNA genes (37), and rRNA genes (8). The GC content was highly conserved (36.6%). However, there was significant diversity in the GC content among the different regions of the cp genome. The GC content in the IR region was significantly higher than that in other regions, which may be due to the distribution of rRNA genes.

Moreover, the GC content of the rRNA gene is more abundant. Previous studies have shown that the IR region plays a significant role in preserving essential genes and stabilizing cp DNA structure (*Palmer & Thompson, 1982*; *Asaf et al., 2018*). In general, the change of cp genome length is due to the contraction and expansion of the IR region, which is also a common evolutionary phenomenon in plants (*Wang et al., 2008*). In our study, the length of the chloroplast genome of Red Fuji in different regions was conserved (26,307–26,308 bp). The gene distribution in the border regions was similar, with no apparent changes. Gene deletion and gene rearrangement often occur in the genome of terrestrial plants (*Knox & Palmer, 1998*). However the cp genome of Red Fuji is highly conserved, and its

characteristics are consistent with the slow evolution of the cp genome of terrestrial plants (*Rivas, 2002*).

As the linker of nucleic acids and proteins, codons play a significant role in genetic information transmission (*Zhang et al., 2019*). Codon use biased genetic information carrier molecular DNA translation is related to the synthesis of biological functional molecular proteins, which is of great significance (*Quax et al., 2015*). However, in the 24 cp Red Fuji genomes sequenced, there was no difference in codon usage, and the most used amino acids were leucine, serine, and arginine. Codons containing A and T bases, in which the third ended with A and T, were preferred. This result is consistent with the previous analysis of codon preference in the cp genome in the Rosaceae (*Liu, He & Qiu, 2021*).

Previous research has suggested that repetitive sequences significantly influence the sequence differences of the cp genome (*Bausher et al., 2006*). We identified 1,176 repeats, each of which had 49 repeats, the most common of which were positive repeat sequences. Each material detected a consistent number of repeats, which showed almost no diversity between them. Chloroplast SSR loci are often used to identify species and heritage analysis because of their abundant distribution and high polymorphism (*Wenpan et al., 2013*; *Morgante, Hanafey & Powell, 2002*). A total of 1,380 SSR sequences were detected, which were mononucleotide repeats. Most of the cpSSRs were short polyadenine (poly-A) or polythymine (poly-T) repeats. Thus, cpSSR markers developed in the Red Fuji cp genome could evaluate genetic diversity and potentially to distinguish different germplasm (*Yi, Kim & Zhang, 2012*). This study, showed that the repetitive sequences and SSR loci from the nine regions have high similarity, which is also evidence of the low rate of cp genome variation in Red Fuji. The whole sequence alignment of cp genomes between *M. sieversii* and Red Fuji from the nine different regions showed high sequence identity with species in the Rosaceae. Phylogenetic analysis showed the same results. The coding regions were more conserved than the non-coding regions, and the LSC and SSC regions were more divergent from each other than the IR regions. Mutations in the inverted region, are transformed, reducing this region's mutation frequency (*Shaw & Small, 2005*). The nucleotide polymorphism of the 24 materials was 0, and no SNPs were detected. There were only two indel sites in the *rps* 16 and *ndh* A genes. The above results further confirm that the variation rate of cp DNA of Red Fuji apples in different regions was very low. A low mutation rate was also found in *Namrata et al. (2017)* at the somatic level. Other studies on the cp genome level also show high conservation of the cp genome. For example, *Feng et al. (2020)* explored the cp genome structure variation of wild and cultivated Qak, and found that gene number and gene structure were almost identical, demonstrating that the cp genome is conserved, but cultivated species have more variation sites than wild species. *Muraguri et al. (2020)* also found that the structure and content of the cp genome in castor are conservative.

## CONCLUSIONS

In the present research, the complete cp genome of Red Fuji was assembled *de novo* using Illumina high-throughput sequencing data. The genomic structures of the 24 samples were compared and analyzed. On this basis, we concluded that the cp genome structure and

gene content of Red Fuji apples in different regions showed little difference. Sequence alignment showed almost no variation in the chloroplast DNA sequence of Red Fuji apples in the different areas. These results are consistent with previous studies reporting slow cp genome evolution. These findings provide a theoretical basis and technical support for the genetic breeding of apples.

### Funding

This study was supported by the Open project of Xinjiang Key Laboratory of Biological Resources and Genetic Engineering (grant 2020D04033 to Xinmin Tian) Tian) and the National Natural Science Foundation of China (grant 31760102 to Xinmin). The funders had no role in study design, data collection and analysis, decision to publish, or preparation of the manuscript.

### Grant Disclosures

The following grant information was disclosed by the authors:
The Open project of Xinjiang Key Laboratory of Biological Resources and Genetic Engineering: 2020D04033.
The National Natural Science Foundation of China: 31760102.

### Competing Interests

The authors declare there are no competing interests.

### Author Contributions

- Haoyu Miao performed the experiments, analyzed the data, prepared figures and/or tables, authored or reviewed drafts of the paper, and approved the final draft.
- Jinbo Bao analyzed the data, prepared figures and/or tables, authored or reviewed drafts of the paper, and approved the final draft.
- Xueli Li and Zhijie Ding analyzed the data, authored or reviewed drafts of the paper, and approved the final draft.
- Xinmin Tian conceived and designed the experiments, authored or reviewed drafts of the paper, and approved the final draft.

### DNA Deposition

The following information was supplied regarding the deposition of DNA sequences:
The data is available at GenBank: OK514184 to OK514191, OK585094 to OK585108 and OK458680.

### Data Availability

The chloroplast genome annotation files of 24 Red Fuji are available in the Supplemental Files.

## Supplemental Information

Supplemental information for this article can be found online at http://dx.doi.org/10.7717/peerj.12927#supplemental-information.

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
