# Peer review of "Comparative analyses of chloroplast genomes in ‘Red Fuji’ apples: low rate of chloroplast genome mutations"

_PeerJ, doi:10.7717/peerj.12927_

## Round 0.1 · original submission · Major Revisions

Dear Dr. Xinmin Tian and co-authors,

As you will see, all the reviewers found the manuscript as potentially interesting. However, more efforts need to make to bring the manuscript to the standard of PeerJ.

Reviewer 1 pointed out that the purpose of using the 24 samples of Red fuji was not clear. Reviewer 2 had concerns about the method description. For example, lacking of description of the whole process from the DNA extraction to the final analysis. Reviewer 3 raised some issues related to experimental reproducibility and the data transparency. I also think that the whole manuscript needs extensive language editing.

I would like to ask you to address or to respond with reasons not to follow the suggestion made by these reviewers.

Best regards,
Atsushi Fukushima

Reviewer 1 ·

Basic reporting

The manuscript by Miao and colleagues report on the determination of chloroplast genomes from 24 Red fuji with detailed analysis of genetic variants study from comparative analysis of chloroplast genomes. In fact, chloroplast genomes at the same species level are generally not different. Although the findings of the authors are very clear, the purpose of the comparative analysis with the chloroplast genome analysis of 24 samples of the same species should be clarified. In a way, this result can be a natural result. Considering the result of producing 50 Gb of data, I thought that it may have been carried out as a previous study for population genetic research such as GBS, GWAS for Red fuji. Please state clearly the purpose of using the 24 samples of Red fuji.

Experimental design

It is very clear except for the material part.

Validity of the findings

It should be noted that the 24 chloroplast genomes are highly conserved.

Annotated reviews are not available for download in order to protect the identity of reviewers who chose to remain anonymous.

Reviewer 2 ·

Basic reporting

There are numerous language error in the manuscript. I would suggest that the author send the manuscript for a proofread by a native speaker.

Some of the figure were not clear, in particular Figure 5 the annotation for tRNA was not clear.

The author should indicate the published apple chloroplast genome by Yan et al. (2019) in the introduction.

Experimental design

Author showed lack of analysis for the chloroplast that were sequenced. There were also lack of description for de novo assembly of the chloroplast and information about the reads obtained from sequencing. Details on the comment for methods can be find in attachment. Author also did not mention that the reads or the sequence assembled were deposited to the public database. Method for DNA extraction is also lacking and the sequencing platform used is also not mentioned.
Author should also include a comparison with the published apple chloroplast as reference as it is already available.
Research question is well defined but the analysis done is not enough to answer the research question. Phylogenetic analysis was lacking and the description to the variation seen across the 24 chloroplast sequenced were also not detailed in manuscript.

I also have a concern over the correction done at the start and stop codon as indicated by the author. What is the exact correction that was done to determine the start and stop codon?

Validity of the findings

The author certainly has develop a good research question but the analysis done was not enough to answer the question.
The author has sequenced 24 chloroplast but in analysis only include 9 chloroplast for comparison by localities, so how to justify variation that may exist for sample from same localities. Some of the literature used to support argument are also very old (eg line 230).

Additional comments

The research has a merit but the analysis done is lacking and still has a room for improvement. Author can further improved in term of phylogenetic analysis and to describe in detail the variation exist among the 24 chloroplast sequenced.

Annotated reviews are not available for download in order to protect the identity of reviewers who chose to remain anonymous.

·

Basic reporting

Miao et al. sequenced and characterized the entire chloroplast genomes of 24 Fuji apple cultivar 'accessions.' The article is written in a standard format similar to other published articles, and the science is sound and easily understandable, though it could be improved further, as indicated in the comments below.

For a strong basis for the rationale of this study, consider introducing what has been done in terms of chloroplast genome sequencing in Malus, with a special emphasis on cultivated apples such as M. domestica and M. pumila. Although the topic of this article appears to be confined to within cultivar characterization and comparison (identification of mutation hotspot areas), you can use previously reported cp genomes to provide insights into the evolution of the cp genome in cultivated species/cultivars. Also, it is not clear why you used the cp genomes of M. sierversii and M. sylvestri in the various analyses for comparison, and why you excluded the cultivated species. The language requires thorough polishing to enhance clarity; see a few examples below.

Specific comments:
Language.
Line 20: ‘…, which belongs to…’
Line 24: what materials? You may consider using terms like varieties/accessions/cultivar…
Line 39: “…fruits (Jung, S & D, Main, 2014). In 2018, China, which is the largest producer of apples in the world (Li et al., 2019), produced more than half (43.88 million tons) of the global production (80.82 million tons).
Line 45: such as
Line 48: how is ‘colorfulness’ an advantage?
Line 50: ‘…longer shelf-life…’
Line 51 delete ‘by consumers’
Line 52: …on genetic…
Line 56 …organelle…
Line 57… commonly used
Line 71: In recent years, the number of…
Line 77: …not only deal with…
Line 81: ‘climate…’, ‘…not yet known…
Line86: ‘…to investigate mutations in the chloroplast DNA of the Red Fuji apple.’
Line 91: ‘…was collected from nine regions in China.’
Line 93: ‘…taken to the laboratory.’
Line 96’…using second-…
Line 100: ‘…to correct…’
Line 218: ‘…was rarely low’ this is somewhat ambiguous as it can be interpreted to mean that the variation rate was always high?

The following corrections can significantly improve
Consider stating the results without repeating details that are already presented in the methodology section. For example, you can delete the first sentence in line 194; ‘Using the MISA software… in line 199; line 207-208; line 214-215 restructure the sentences to remove the software used.
Line 139: delete ‘collected’
Line 139-140: what about the structure of the cp genome?
Line 157: Give details in the materials and method section on how the nine samples were selected. Was it random?
Discussion
Lines 270-273: which study is relevant here Li et al’s or Fang et al’s, both please rewrite the sentence and consider adding the year for Li et al’s study.

Experimental design

The study's objectives are in line with the journal's scope.
The methods require further input to allow for reproducibility. See comments below.

Give details on how the DNA was extracted, how quality was checked
How data cleaning was attained, and using which programs/parameters?
What was the amount of clean data for each sample?
Line 98: provide the version and reference for each of the software/program/website-based tools used.
Line 104: please consider writing all abbreviations in full at first mention…
Line 126: expound on the motif settings to allow for reproducibility

Validity of the findings

There is no information on whether and where the data was deposited. The data could not be reviewed and verified.

Conclusion:
Lines 280-281: This conclusion is not supported in any way by the performed analyses. How can you explain that timeline?

---

## Round 0.2 · accepted · Accept

Dear authors,

Thank you for revising.

Best regards

Reviewer 1 ·

Basic reporting

The revised manuscript has been properly revised for publication.

Experimental design

It has been properly prepared for publication in this journal.

Validity of the findings

It has been properly prepared for publication in this journal.